# Interventions in sexual and reproductive health services addressing violence against women in low-income and middle-income countries: a mixed-methods systematic review

Natalia V Lewis ,[1] Muzrif Munas,[1,2] Manuela Colombini,[3] A F d'Oliveira,[4] Stephanie Pereira,[4] Satya Shrestha,[1,5] Thilini Rajapakse,[2] Amira Shaheen,[6] Poonam Rishal,[5] Abdulsalam Alkaiyat,[6] Alison Richards,[1,7] Claudia M Garcia-Moreno,[8] Gene S Feder ,[1] Loraine J Bacchus [3]

For numbered affiliations see end of article.

**Correspondence to**
Dr Natalia V Lewis;
nat.lewis@bristol.ac.uk

## ABSTRACT

**Objectives** To synthesise evidence on the effectiveness, cost-effectiveness and barriers to responding to violence against women (VAW) in sexual and reproductive health (SRH) services in low/middle-income countries (LMICs).

**Design** Mixed-methods systematic review.

**Data sources** Medline, Embase, Psycinfo, Cochrane, Cinahl, IMEMR, Web of Science, Popline, Lilacs, WHO RHL, ClinicalTrials.gov, Google, Google Scholar, websites of key organisations through December 2019.

**Eligibility criteria** Studies of any design that evaluated VAW interventions in SRH services in LMICs.

**Data extraction and synthesis** Concurrent narrative quantitative and thematic qualitative syntheses, integration through line of argument and mapping onto a logic model. Two reviewers extracted data and appraised quality.

**Results** 26 studies of varied interventions using heterogeneous outcomes. Of ten interventions that strengthened health systems capacity to respond to VAW during routine SRH consultation, three reported no harm and reduction in some types of violence. Of nine interventions that strengthened health systems and communities' capacity to respond to VAW, three reported conflicting effects on re-exposure to some types of VAW and mixed effect on SRH. The interventions increased identification of VAW but had no effect on the provision (75%–100%) and uptake (0.6%–53%) of referrals to VAW services. Of seven psychosocial interventions in addition to SRH consultation that strengthened women's readiness to address VAW, four reduced re-exposure to some types of VAW and improved health. Factors that disrupted the pathway to better outcomes included accepting attitudes towards VAW, fear of consequences and limited readiness of the society, health systems and individuals. No study evaluated cost-effectiveness.

**Conclusions** Some VAW interventions in SRH services reduced re-exposure to some types of VAW and improved some health outcomes in single studies. Future interventions should strengthen capacity to address VAW across health systems, communities and individual

### Strengths and limitations of this study

► This review was carried out by a team of researchers from the UK and low-income and middle-income countries with expertise and experience in health system responses to violence against women and global health.
► Inclusion of peer-reviewed and grey reports of studies of any design resulted in selection of the most relevant studies.
► The logic model approach to the integration of synthesis findings produced evidence in a format understandable to the end-users of this review.
► Most included studies had methodological limitations and high risk of bias.
► We could not perform meta-analysis of quantitative findings because primary studies evaluated varied interventions and used different instruments to measure varied outcomes.

women. First-line support should be better tailored to women's needs and expectations.

**PROSPERO registration number** CRD42019137167.

## INTRODUCTION

Violence against women (VAW) is a violation of global health and human rights.[1] The most common forms of VAW are intimate partner violence (IPV) and non-partner sexual violence. One in three women worldwide have experienced physical and/or sexual violence, mostly by an intimate partner. VAW is more prevalent in low/middle-income countries (LMICs). Exposure to VAW is associated with mental and physical health problems, including increased sexually transmitted infection and HIV, unplanned pregnancy and abortion, gynaecological

conditions.[2 3] Although IPV against men is increasingly recognised within the context of both same sex and heterosexual relationships, the phenomenon of male victimisation and its health consequences is still poorly understood.[4] There is a dearth of primary research on the healthcare response to male victims and perpetrators.[5]

The healthcare system has a key role in preventing VAW because most women attend sexual and reproductive health (SRH) services at some point.[6 7] The main role of the healthcare system is to contribute towards secondary and tertiary prevention through early detection of VAW and mitigation of its impact which can prevent ill health and reoccurrence of violence. Healthcare providers (HCPs) are uniquely placed to identify victims/survivors, provide first-line support and clinical care, and connect them with other services. Healthcare systems can also contribute to primary prevention through early identification of children exposed to violence in the home and support to programmes like home visiting or early childhood development.[8] The capacity of healthcare systems to respond to VAW is defined as the cumulative availability and strength of the following *building blocks* from the Health Systems Wheel: (i) leadership and governance, (ii) multi-sectoral coordination, (iii) workforce development, (iv) healthcare delivery, (v) infrastructure, (vi) financing, (vii) monitoring and evaluation (WHO 2010).[9 10] The Health Systems Wheel[11] highlights key components that need to be in place to support individual HCPs and organisations to offer a comprehensive and client-centred response to VAW. It assumes that all elements of the health system—individual, organisational, contextual and structural—impact on provision of response to VAW. The WHO guidelines for evidence-based health systems response to VAW adopted the Health Systems Wheel framework to recommend intervention activities across the health systems *building blocks*.[12 13] In LMICs, healthcare delivery for VAW has been implemented through integration at the level of individual HCPs, healthcare facility, and healthcare system.[14]

Systematic reviews[15 16] and WHO guidelines[17] found scant evidence from LMICs on effectiveness of VAW interventions in healthcare. This study addresses the gap by answering two questions: (i) what is the evidence for effectiveness and cost-effectiveness of interventions in SRH services that address VAW? (ii) what are the barriers to the effectiveness?

## METHODS

We conducted concurrent quantitative and qualitative evidence syntheses with integration into a line of argument[18] and mapping onto a logic model.[19] The mixed-methods design allowed integration of diverse types of evidence to inform VAW research and intervention development in LMICs. Our analysis was informed by the WHO Health Systems Wheel framework for responding to VAW.[8] We defined the health systems capacity to respond to VAW as the cumulative availability and strength of the Health Systems Wheel *building blocks*. We looked at the capacity of the health systems at three levels: individual providers (eg, attitudes, knowledge, confidence, behaviour and practices), services and organisations (eg, infrastructure, availability of supplies/medicines), community (attitudes, knowledge, practices).[13] We defined women's capacity to respond to VAW as their readiness and ability to seek help, disclose abuse, get referrals and receive services. We followed the Cochrane[20] and Preferred Reporting Items for Systematic Reviews and Meta-Analyses guidelines.[21]

### Search strategy and selection criteria

We included primary intervention studies reported in any language with an English abstract published since 2005, the year of the first published evaluation of VAW interventions in SRH services (expert opinion from the study advisory group). We identified earlier studies through reference checking. We used terminology and definitions from WHO guidance on strengthening health systems to respond to VAW (table 1).[13]

An information specialist (AR) applied the search strategy to Medline, Embase, Psycinfo, Cochrane, Cinahl, IMEMR, Web of Science, Popline, Lilacs, WHO RHL, ClinicalTrials.gov (20 August 2018 and 3–4 December 2019) (online supplemental file 1). AR searched for grey literature via Google, Google Scholar and websites of key organisations in the field of VAW and SRH in LMICs (UNFPA, SVRI, JPHIEGO, USAID, WHO (IRIS) SEARO, WHO (IRIS) EMRO, World Bank). AR uploaded all records into EndNote and deduplicated. Two pairs of reviewers (NVL and MM, AFD and MC) independently assessed eligibility. Disagreements were resolved through consensus or third opinion (LB). NVL checked references and citations.

### Data analysis

NVL adapted the Cochrane Effective Practice and Organisation of Care (EPOC) data extraction form.[22] We collated multiple reports from the same study and used the most detailed report as the primary source for extracting study results. The included studies were divided among reviewers who worked in pairs, one to extract data and another to check. The pairs reconciled data extraction through discussion. We extracted study details on setting, study design and aim, sample size, participants characteristics, intervention characteristics and theories, and outcomes relevant to our review questions. For each quantitative outcome, we extracted type of measure and effect estimates as reported in the primary study. If authors did not report intervention effects, we extracted the postintervention point estimate. If a follow-up measure was reported repeatedly, we extracted the latest measure. We judged intervention effectiveness by improvement in any primary or secondary outcome listed in the individual studies (table 1). We used authors' interpretation of their findings based on statistical significance or 95% CIs and categorised effect estimates as *improvement, mixed effect* or *null effect*. We ascribed a *mixed effect* when one or more, but

**Table 1** Study inclusion and exclusion criteria with justification

| | Inclusion criteria | Exclusion criteria |
|---|---|---|
| **P**articipants | Recipients of healthcare services—women of reproductive age (15–49 years old) AND/OR Healthcare providers—organisations (eg, hospital, clinic, primary care centre, other service delivery points) or individuals (eg, healthcare professional, community health worker or any other person who is trained to deliver healthcare in their community). Studies which recruited only a subset of recipients or providers of healthcare services. | Female children and girls under 15 years old. While recognising that pregnancies occur among young adolescents 10–14, most studies consider women aged 15–49 years as the main group using SRH services in LMICs. Male recipients of healthcare services. |
| **I**nterventions | Any intervention addressing violence against women (VAW). These are complex interventions aimed to identify women affected by violence, provide first-line support, clinical care, and signpost, or refer to available community support services including specialist VAW services. Any definition of VAW, including any type of IPV, domestic violence and abuse, family violence or non-partner sexual violence against a woman, including transgender women. | No intervention Hypothetical intervention addressing VAW. We are synthesising evidence of interventions that have been tested. Female genital mutilation/cutting, trafficking. These types of VAW were addressed in recent systematic reviews. 'Honour'-based violence, forced marriage. There is an overlap between IPV, domestic violence and abuse and 'honour'-based violence and forced marriage. Therefore, we will capture relevant studies through including papers on IPV and domestic violence and abuse. |
| **C**omparators | Controlled studies: usual care, no VAW intervention, delayed VAW intervention, minimal intervention (eg, information provision). Uncontrolled studies: group before the intervention. No control group. | |
| **O**utcomes | Outcome is an event or measurement collected for participants in a study. Primary outcomes: any health outcomes for survivors of VAW (for example, re-exposure to VAW, sexual and reproductive health, mental health, physical healthy, quality of life), any harms, cost-effectiveness of VAW interventions. AND/OR Secondary outcomes: patient and provider health-related cognitive and emotional outcomes (eg, knowledge, attitudes, confidence, readiness); health-related behaviour and practices (eg, identification and disclosure of VAW, provision and uptake of referrals and SRH services). Phenomenon of interest: provider and recipient experiences of and views on VAW interventions. | |
| **S**tudy type | Primary intervention studies of any design. Primary studies that used quantitative designs such as randomised controlled trials, controlled and uncontrolled before-after studies, interrupted time series studies, cross-sectional studies. Primary studies that used qualitative designs such as ethnographic research, interview or focus-group based studies, case studies, process evaluations and mixed methods designs. We include these studies if they had used qualitative methods for data collection and analysis and reported quotes from participants. Mixed-methods studies. | Systematic reviews. We used systematic reviews to identify potentially eligible primary studies. |
| Context | Studies conducted in SRH services in a country defined as LMIC by the World Bank, including humanitarian settings. Depending on country context, SRH services can be delivered at any level of healthcare provision and usually include contraceptive services, maternal and perinatal health, treatment for STI, HIV and reproductive tract infections, abortion, fertility treatment and gynaecological treatment. | |
| Report type | Full-text peer-reviewed studies, conference abstracts, grey literature, unpublished studies. | Animal studies, opinion pieces, editorials and publication which did not report primary data. |

HIV, human immunodeficiency virus; IPV, intimate partner violence; LMICs, low-income and middle-income countries; SRH, sexual and reproductive health; STI, sexually transmitted infection; VAW, violence against women.

not all measures of the same outcome changed under the same intervention (eg, reduction in physical and sexual but not psychological IPV, improvement in some coping behaviours but not in others). NVL asked corresponding authors to check data extraction forms for their studies and provide missing information; nine responded.

Reviewers assessed the quality of the primary studies as part of data extraction. For randomised controlled trials (RCTs), we used the Revised Cochrane risk-of-bias tool for randomised trials.[23] For quasi-experimental studies we adapted the criteria listed by the EPOC Group.[24] For qualitative studies we adapted the Critical Appraisal Skills Programme (CASP) Qualitative Checklist.[25] We did not exclude studies based on their methodological quality.

We summarised interventions by mapping them onto the Health Systems Wheel[8] and models of health system responses to VAW in LMICs.[14] It was not possible to conduct a meta-analysis of quantitative outcomes due to the heterogeneity of the interventions, the outcomes, and their measurement. We undertook a narrative quantitative synthesis[26] and thematic qualitative synthesis,[27] summarised quantitative and qualitative syntheses in tables, and integrated them through a line of argument[18] and mapping onto a process-oriented logic model.[19] Reviewers (NVL, MC, LB) drafted the logic model by mapping primary and secondary outcomes in the hypothesised logical order of occurrence and refined it through three iterative cycles of revisions.

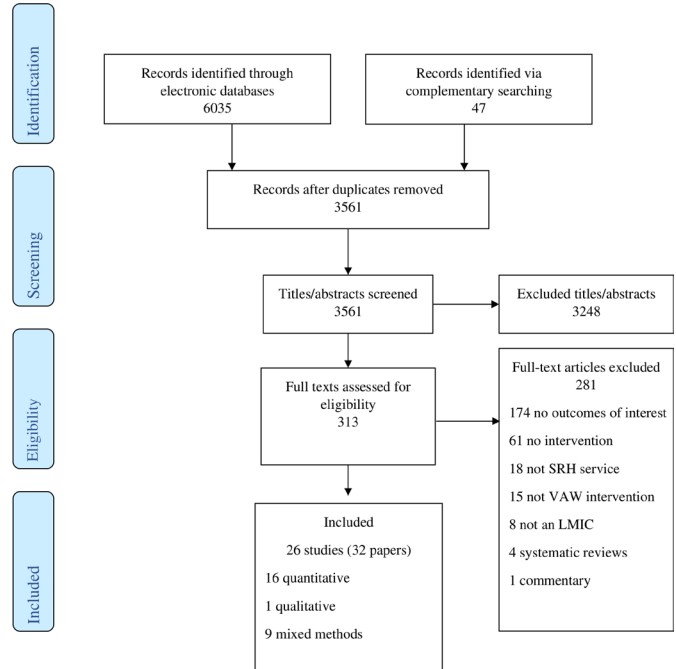

**Figure 1** Flow diagram. LMICs, low/middle-income countries; SRH, sexual and reproductive health; VAW, violence against women.

1. Direct effects result from intervention activities producing structural changes at service level and changes in health-related cognitive and emotional outcomes among HCPs and women indicating improvement in the health systems capacity and women readiness to respond to VAW.
2. Intermediate effects result from direct effects producing changes in health-related behaviour and practices indicating improvement in the health systems capacity and women readiness to respond to VAW. HCPs identify women affected by VAW and provide first-line support; women disclose VAW, use offered support, develop adaptive coping strategies.
3. Health outcomes result from intermediate effects producing changes in women health and safety indicating improvement in their readiness to cope with VAW. Women use adaptive coping strategies and safety behaviours; these lead to reduction in re-exposure to VAW and better health.

## Patient and public involvement

No patients or members of the public were involved in this study.

## RESULTS

Searches identified 6082 records, we assessed 313 full text reports and included 32[28–59] reporting on 26 studies[28–35 38–40 42 45–47 49 50 52–60] (figure 1, online supplemental file 2).

## Characteristics of included studies

Of the 26 studies, 18 were from sub-Saharan Africa,[28 31–34 38 42 45–47 49 52–54 56 57 59 60] 3 from the Middle East,[40 55 58] 3 from South Asia[29 39 50] and 2 from South America.[30 35] Twelve quantitative evaluations were RCTs,[32 34 35 38 40 47 50 52 55 58–60] six were uncontrolled before-after (UBA) studies,[29 30 39 42 46 54] six cross-sectional studies[31 45 49 53 56 57] and one was a controlled before-after evaluation.[28] Nine qualitative evaluations were components of mixed-methods studies: three embedded in RCTs,[38 44 51] two carried out alongside UBA studies,[30 54] three alongside cross-sectional studies[49 56 57] and one stand-alone qualitative study.[33] No two studies of similar design evaluated the same intervention and outcomes. The duration of follow-up period ranged from two weeks[33] to 4 years and 7 months.[59]

Most interventions took place in antenatal care (ANC) services (n=11),[29 34 35 39 40 45 47 50 55 56 58] followed by HIV testing and treatment (n=8),[32 33 38 46 52 57 59 60] services for victims of sexual violence (n=5)[28 31 42 53 54] and family planning (n=2).[30 49] The SRH services were provided in primary care (n=15),[30 32–34 39 40 46 47 49 52 53 55 56 58 59] hospital (n=7)[29 35 38 42 45 50 57] and across both (n=4).[28 31 54 60] Included studies used different definitions and measures of VAW. A majority (n=11) targeted IPV.[32 33 35 38 39 45 46 49 55 57 59] Six interventions targeted sexual violence by intimate partners and non-partners.[28 31 42 52–54] Five interventions focused on domestic violence (DV) from any family member[29 34 40 50 58] and four targeted VAW from intimate partners and non-partners.[30 47 56 60] In studies that reported sample size, 901 HCPs received VAW interventions with the average sample size of 100 ranging from 4[49] to 408.[39] A total of 12 078 women of reproductive age received VAW interventions, with the average sample size of 549 ranging from 32[52] to 2081.[31]

## Quality appraisal

Most quantitative studies were at high risk of bias (online supplemental file 3). Of 12 RCTs, seven had high risk of bias from deviations in intervention adherence,[32 38 40 47 58–60] 7 had high risk from measurement of outcomes[35 38 40 47 52 59 60] and 7 had concerns from selective reporting of outcomes.[35 38 47 52 55 58 59] Of 13 non-randomised studies (all at high risk of bias), only 4 adequately addressed missing outcome data.[28 29 49 54] Of 10 qualitative evaluations, 5 scored 15 and above on the 20-point CASP checklist,[44 45 49 51 56] indicating relatively high quality of research design and conduct. The main weaknesses were insufficient justification of methods, reporting of recruitment and strategies for neutrality.

## Types of interventions

All interventions were complex healthcare interventions,[61] however, only two[30 60] included components across all domains on the Health Systems Wheel[8] (table 2) and only four were theoretically informed.[32 34 50 59]

Most VAW work was delivered by a single HCP (n=10) or by several HCPs within the same facility (n=9). Only six

**Table 2** Included interventions mapped on the Health Systems Wheel framework and models of service integration

| Study ID | Study design | Leadership and governance | Multi-sectoral coordination | Workforce development | Healthcare delivery | Infrastructure | Financing | Information | Level of VAW service integration |
|---|---|---|---|---|---|---|---|---|---|
| Abeid et al[28] | CBA | | | ● | ● | ● | ● | ● | Systems |
| Arora et al[29] | UBA | | | | ● | | ● | | Provider |
| Bott et al[30] | UBA | ● | ● | ● | ● | ● | ● | ● | Facility |
| Bress et al[31] | Cross-sectional | | ● | ● | ● | ● | ● | ● | Provider |
| Brown and Van Zyl[32] | RCT | | | | ● | | ● | | Facility |
| Cockcroft et al[34] | cRCT | | | ● | ● | ● | ● | ● | Provider |
| Cripe et al[35] | RCT | | | ● | ● | | ● | | Provider |
| Christofides and Jewkes[33] | Qualitative | | | ● | ● | | ● | | Facility |
| Haberland et al[38] | RCT | | | ● | ● | ● | ● | ● | Facility |
| Jayatilleke et al[39] | UBA | | | ● | ● | | ● | | Provider |
| Khalili et al[40] | RCT | | | | ● | | ● | | Provider |
| Kim et al[42] | UBA | ● | | ● | ● | ● | ● | ● | Facility |
| Laisser et al[45] | Cross-sectional | | | ● | ● | | ● | ● | Systems |
| Matseke and Peltzer[46] | UBA | | | ● | ● | | ● | | Systems |
| Mutisya et al[47] | RCT | | | | ● | | ● | | Provider |
| Samandari et al[49] | Cross-sectional | ● | ● | ● | ● | ● | ● | ● | Systems |
| Sapkota et al[50] | RCT | | | | ● | | ● | | Provider |
| Settergren et al[60] | cRCT | ● | ● | ● | ● | ● | ● | ● | Systems |
| Sikkema et al[52] | RCT | | | ● | ● | | ● | | Provider |
| Sithole et al[53] | Cross-sectional | | ● | ● | ● | ● | ● | ● | Facility |
| Smith et al[54] | UBA | | | ● | | | ● | | Facility |
| Taghizadeh et al[55] | RCT | | | | ● | | ● | | Provider |
| Turan et al[56] | Cross-sectional | | ● | ● | ● | ● | ● | ● | Systems |
| Undie et al[57] | Cross-sectional | | | ● | ● | | ● | ● | Facility |
| Vakily et al[58] | RCT | | | ● | | | ● | | Facility |
| Wagman et al[59] | cRCT | ● | ● | ● | | | ● | | Facility |

Provider-level integration when one trained healthcare provider (HCP) delivers most of the VAW work. Facility-level integration when several trained HCPs deliver most VAW work within one healthcare facility. Systems-level integration when trained HCP identifies patients affected by VAW, provides first-line support and clinical care, and then refers them to higher level facilities with VAW specialist or external VAW services.

CBA, controlled before-after; cRCT, cluster randomised controlled trial; RCT, randomised controlled trial; UBA, uncontrolled before-after; VAW, violence against women.

interventions were integrated at a systems-level with HCPs identifying VAW cases, providing clinical care and first-line support, and referring to external VAW services.[28 45 46 49 56 60] We clustered 26 interventions into three categories based on the target group(s) and location of the common activities (online supplemental file 2).

### Response to VAW during routine SRH consultation (n=10)

These interventions aimed to strengthen health system capacity to respond to VAW through integrating identification and first-line support into routine SRH consultations.[28 32 33 38 39 45 46 57 58] This comprised training for HCPs in VAW screening, basic psychosocial counselling, and linkage to VAW resources. Training aimed to improve HCP knowledge, attitudes and practices on VAW. Identification and response by trained HCPs aimed to increase women's readiness to respond to VAW. Duration of the integrated SRH-VAW consultation ranged between 7[32] and 30 min.[38 46]

### Response to VAW during routine SRH consultation plus community engagement (n=9)

These interventions aimed to strengthen health system capacity to respond to VAW across SRH service and surrounding communities.[30 31 34 42 49 53 56 59 60] Service-based activities were similar to the first category. The community-based activities aimed to shift gender norms and improve access to integrated SRH-VAW services through raising awareness about post-rape care,[31 42 53]

education on gender and VAW[30 49 53 56 59 60] and couples' education about VAW.[34 60] Integrated SRH-VAW consultations supported by community engagement aimed to increase women's readiness to respond to VAW.

### Response to VAW in addition to routine SRH consultation (n=7)

These interventions aimed to strengthen women's readiness to respond to VAW.[29 35 40 47 50 52 55] Study personnel screened women attending routine SRH services and delivered the interventions to self-selected women with experience of VAW. This comprised more intensive support through specialist psychosocial counselling,[29 35 47 50] coping skills training[52 55] and psychoeducation.[40 52 55] The average number of sessions was three (range 1–7) with each session lasting from 30[29 35 47] to 90 min.[40 52 55] Interventions were delivered face-to-face individually,[29 35 40 47 50] in a group[55] and mixed format.[52]

### Interventions effects and outcomes

The logic model displays all outcomes of interest in the three intervention categories (figure 2). The arrows illustrate the hypothesised flow of change from intervention activities through health-related direct and intermediate effects to health outcomes.

None of the primary studies reported outcomes at service level. Most studies that evaluated interventions that aimed to strengthen health system capacity to respond to VAW across SRH service and community measured direct and intermediate effects on HCP and women's knowledge, attitudes and behaviour. In contrast, all studies that evaluated interventions that aimed to strengthen women's readiness to cope with VAW reported their health outcomes, but only a few looked at preceding changes in women's cognition, emotions and behaviour (figure 2, table 3).

### Direct effects on cognition and emotions

The routine SRH-VAW intervention category had overall positive direct effects on HCP and women's knowledge, attitudes and readiness. Interventions with community engagement reported mixed and improved direct effects.

### Intermediate effects on behaviour and practices

Changes in professional behaviour were measured through the rates of VAW screening, provision of referrals to support services and post-rape care. Changes in women's behaviour were measured through VAW disclosure, uptake of referrals and other services. The overall evidence was uncertain across all three intervention categories with RCTs and non-randomised evaluations reporting improved, mixed and null effects.

### Health outcomes

Only half the studies reported measures of health and re-exposure to VAW; two interventions reported no harm resulting from taking part in interventions,[32 38] and ten led to some health improvement. The overall direction of effect on any outcomes of interest was towards improvement in the *routine SRH-VAW consultation* category and in

the *additional response* category. In contrast, most interventions in the *plus community engagement* category reported mixed or null effect on women's health and re-exposure to VAW. We found that although some interventions did not reduce re-exposure to VAW, none reported violence escalation. Of ten studies that measured re-exposure to VAW, six found a reduction,[29 32 34 40 46 47] two reductions in some violence types but not in others[55 59] and two reported no change.[38 60]

Of 26 studies, only two reported changes across all three domains of the logic model, one from the *routine SRH-VAW consultation* category[38] and one from the *plus community engagement* category.[60] Four evaluations of the *additional response* category reported changes across two domains - intermediate effects on behaviour and practices and women's health outcomes.[29 35 50 52] These six studies were consistent with our hypotheses. If intervention improved women's safety behaviour and use of support services, their health improved.[50] Mixed or null effect on HCP and women's cognitive and behavioural outcomes suggested some explanation for no change in re-exposure to VAW.[38 60] Contradicting direct and indirect effects and outcomes[29 52] indicated possible barriers on the pathway from intervention activities to outcomes.

### Response to VAW during routine SRH consultation

Of ten evaluations, two RCTs[32 38] and one UBA[46] studies reported conflicting findings on re-exposure to VAW; none measured women's health (figure 2, table 3, online supplemental file 4). These three interventions did not lead to escalation of violence. There was some evidence for the reduction in HIV-disclosure-related violence at up to 2-month follow-up[32] and risk of becoming a victim of femicide at 3-month follow-up[46] possibly through some improvement in HCP's and women's cognition and practice. Two RCTs reported that integrated HIV-IPV consultation caused no harm to women.[32 38] However, all studies were at high risk of bias.

An RCT[38] and UBA[57] in the Kenyan hospital with on-site GBV centre reported convergent findings. The UBA study of an integrated HIV-IPV consultation with assisted referral to GBV centre reported 8% IPV disclosure rate, 75% referrals provision and 40% uptake.[57] The RCT of an integrated HIV-IPV consultation with referral to in-service GBV specialist found increased rates of IPV screening but no effect on provision of referrals. The 29 min integrated HIV-IPV consultation increased women's knowledge about VAW and IPV disclosure, but had no effect on their attitudes, readiness to address VAW, uptake of referrals and re-exposure to IPV.[38]

Another RCT of a 7 min integrated HIV-IPV consultation over the phone reported that 62% of women used a safety plan and 80% employed at least one safety strategy, however their use of SRH services and perceived risk and safety did not change. Despite no effect on women's behaviour, the trial reported a fourfold reduction in HIV-disclosure-related violence (OR 4.37; 95% CI 1.46 to 13.44).[35] One UBA found that a 30 min integrated

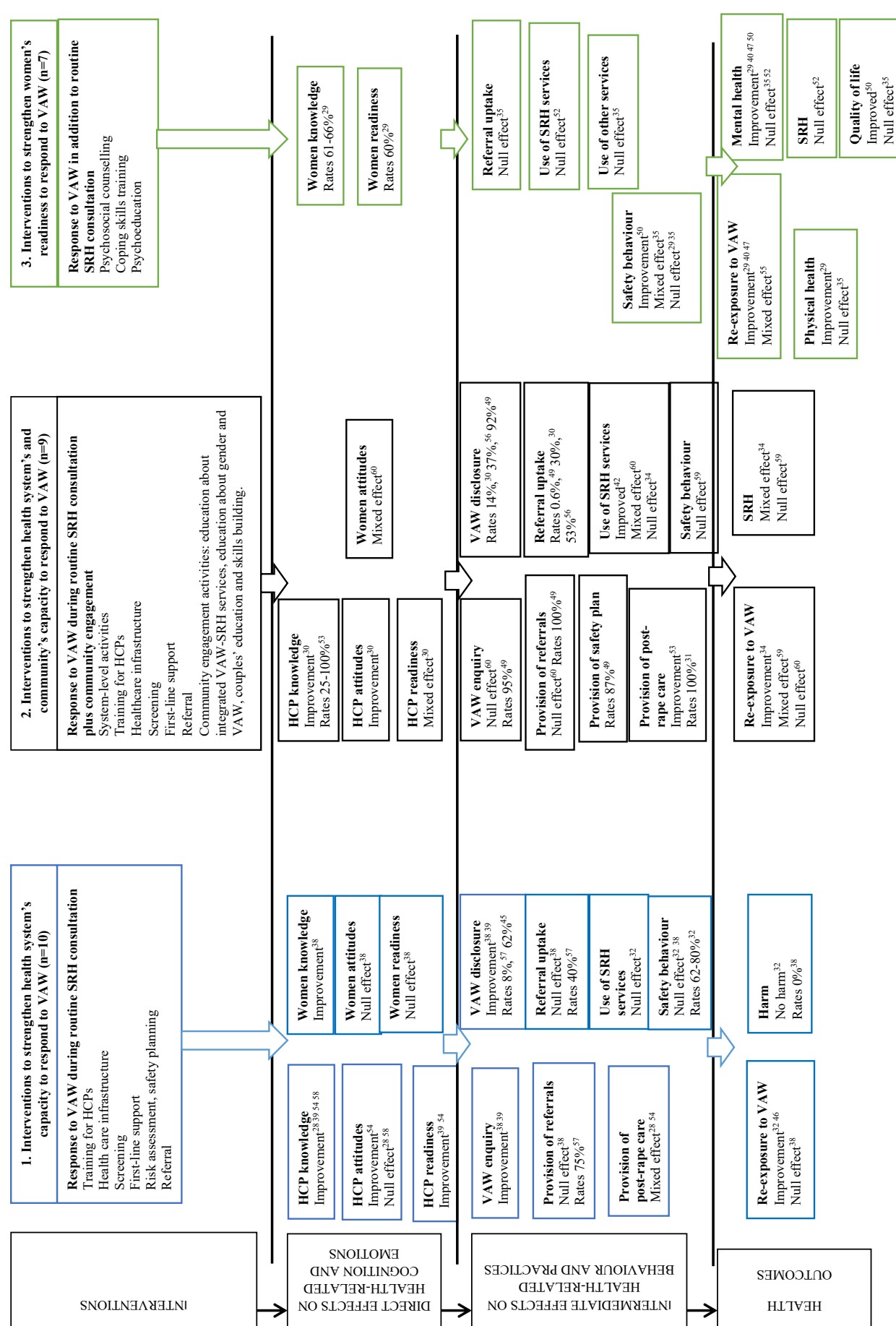

**Figure 2** Process-oriented logic model of interventions in sexual and reproductive health services addressing violence against women in low-income and middle-income countries. HCP, healthcare provider; SRH, sexual and reproductive health; VAW, violence against women.

**Table 3** Health-related effects and outcomes in quantitative randomised and non-randomised evaluations of interventions addressing VAW in SRH services

| Intervention category | Improvement | | Mixed effect | | Null effect | | Studies, n |
|---|---|---|---|---|---|---|---|
| | RCT | Non-randomised | RCT | Non-randomised | RCT | Non-randomised | |
| **Response to VAW during routine SRH consultation (n=10)** | | | | | | | |
| **Direct effect on health-related cognition and emotions** | | | | | | | |
| HCP knowledge | Vakily et al[58] | Jayatilleke et al[39] Smith et al[54] Abeid et al[28] | | | | | 4 |
| HCP attitudes | | Smith et al[54] | | | | Vakily et al[58] Abeid et al[28] | 3 |
| HCP readiness | | Jayatilleke et al[39] Smith et al[54] | | | | | 2 |
| Women's knowledge | Haberland et al[38] | | | | | | 1 |
| Women's attitude | | | | | Haberland et al[38] | | 1 |
| Women's readiness | | | | | Haberland et al[38] | | 1 |
| **Intermediate effects on health-related behaviour and practices** | | Jayatilleke et al[39] | Haberland et al[38] | Smith et al[54] Abeid et al[28] | | | 4 |
| HCP behaviour | | | | | | | |
| Women's behaviour | | | Haberland et al[38] | | Brown and Van Zyl[32] | | 2 |
| **Health outcomes** | | | | | | | |
| Re-exposure to VAW | Brown and Van Zyl[32] | Matseke and Peltzer[46] | | | Haberland et al[38] | | 3 |
| Any harm | | | | | Brown and Van Zyl[32] Haberland et al[38] | | 2 |
| **Response to VAW during SRH consultation plus community engagement** | | | | | | | |
| **Direct effects on health-related cognition and emotions** | | | | | | | |
| HCP attitudes | | Bott et al[30] | | | | | 1 |
| HCP readiness | | | | Bott et al[30] | | | 1 |
| Women attitude | | | Settergren et al[60] | | | | 1 |
| **Intermediate effect on behaviour and practices** | | | | | | | |
| HCP behaviour | | | | | Settergren et al[60] | | 1 |
| Women behaviour | | Kim et al[42] | Settergren et al[60] | | Cockcroft et al[34] Wagman et al[59] | | 4 |

Continued

**Table 3** Continued

| Intervention category | Improvement | | Mixed effect | | Null effect | | Studies, n |
|---|---|---|---|---|---|---|---|
| | RCT | Non-randomised | RCT | Non-randomised | RCT | Non-randomised | |
| **Health outcomes** | | | | | | | |
| Re-exposure to VAW | **Cockcroft et al[34]** | | Wagman et al[59] | | Settergren et al[60] | | 3 |
| Sexual and reproductive health | | | **Cockcroft et al[34]** | | Wagman et al[59] | | 2 |
| **Response to VAW in addition to routine SRH consultation** | | | | | | | |
| Intermediate effects on health-related behaviour and practices | | | | | | | |
| Women behaviour | Sapkota et al[50] | | Sikkema et al[52] | | Cripe et al[35] | Arora et al[29] | 4 |
| **Health outcomes** | | | | | | | |
| Re-exposure to VAW | **Khalili et al[40]** **Mutisya et al[47]** | Arora et al[29] | Taghizaden et al[55] | | | | 4 |
| Sexual and reproductive health | | | | | Sikkema et al[52] | | 1 |
| Physical health | | Arora et al[29] | | | | | 1 |
| Mental health | **Khalili et al[40]** **Mutisya et al[47]** **Sapkota et al[50]** | Arora et al[29] | | | Cripe et al[35] Sikkema et al[52] | | 6 |
| Quality of life | **Sapkota et al[50]** | | | | Cripe et al[35] | | 1 |
| Studies, n | 7 | 7 | 6 | 2 | 7 | 3 | |

Bold indicates studies that reported sample size calculation.

HCP, healthcare providers; RCT, randomised controlled trial; SRH, sexual and reproductive health; VAW, violence against women.

HIV-IPV consultation contributed towards a reduction in the risk of femicide (mean difference 3.2, SD 3.56; 95% CI 2.43 to 3.98).[32]

Other randomised and non-randomised evaluations of varied one-off training for HCPs reported conflicting findings on their knowledge, attitudes and readiness to address VAW. Non-randomised studies reported increased IPV screening rates,[39] low VAW disclosure rates[45] and mixed effect on provision of post-rape care.[28 54]

Qualitative evaluations confirmed that training increased HCP ability to respond to VAW during routine SRH consultations.[38 45 54 57] One evaluation of HCP training on post-rape care described a potential mechanism of impact on HCP negative attitudes by separating personal beliefs about victims from the provision of clinical care.[54] Women found that integrated HIV-IPV consultation improved their knowledge about IPV. They benefited from emotional support and felt empowered.[33 38 57]

### Response to VAW during routine SRH consultation plus community engagement

Three cluster RCTs reported conflicting findings on women's SRH and re-exposure to VAW. The overall effect was uncertain (figure 2, table 3, online supplemental file 5).[34 59 60] A Nigerian RCT at low risk of bias evaluated universal home visits that discussed DV and other risk factors with pregnant women and their spouses. The trial reported no effect on women's use of SRH services, reduction in the proportion who experienced physical DV (RD 0.064 (95% CI 0.045 to 0.084), and mixed effect on pregnancy and birth indicators.[34] A Tanzanian RCT at high risk of bias evaluated integrated HIV-VAW consultation, onsite and external referrals, community and couple education. Intervention had a mixed effect on women's attitudes about VAW and gender roles, no effect on rates of enquiry and referrals, mixed effect on women's use of SRH services and null effect on re-exposure to IPV (OR=0.85, 95% CI 0.62 to 1.16).[60] A Ugandan RCT at high risk of bias evaluated integrated HIV-IPV consultation, onsite referral and community education. The intervention had no effect on women's safety behaviour and null effect on SRH. Re-exposure to physical and sexual IPV reduced (relative prevalence risk ratios (PRR) of 0.74 (95% CI 0.63 to 0.86), 0.75 (95% CI 0.62 to 0.90), respectively), but psychological IPV and HIV did not change.[59]

Non-randomised studies reported more positive effects on HCP knowledge, attitudes, preparedness[30] and provision and use of post-rape care.[31 42 53] They also reported high rates of IPV screening and provision of referrals and clinical care by HCPs vs low uptake of referrals and other services by women.[49 56]

Qualitative evaluations confirmed that VAW training transformed HCP attitudes towards patients and their own work and improved their diagnostic and counselling skills. HCPs appreciated the intervention and expressed a willingness to continue VAW work.[30 49 56 57] Women felt empowered and supported by HCPs.[30 33 45 49] Community engagement raised awareness about SRH-VAW services.[56]

### Response to VAW in addition to routine SRH consultation

This intervention category had the most robust evidence from six RCTs[35 40 47 50 52 55] and one UBA study[29] (figure 2, table 3, online supplemental file 6). The studies reported conflicting results with more evidence for a reduction in re-exposure to VAW at up to 6-month postintervention and improvement in health possibly through improvement in women's safety behaviour. There was some evidence that longer interventions produced better outcomes.[29 40 47]

### Psychosocial counselling

Three RCTs of counselling sessions for pregnant women with experience of VAW reported conflicting results with no evidence for a dose–response.[35 47 50] The overall effect was towards reduction in re-exposure to violence and improvement in health outcomes. An adequately powered Nepalese RCT with low risk of bias evaluated a 35–45 min psychosocial counselling session with a resource card and counsellor's contact details. The trial reported positive effects on women's self-efficacy (MD 0.5; 95% CI 0.30 to 0.72), perceived social support (MD 0.73; 95% CI 0.39 to 1.06), safety behaviours (MD 2.41; 95% CI 1.43 to 3.40), anxiety (MD −3.73; 95% CI −5.42 to −2.04), depression (MD −3.41; 95% CI −4.84 to −1.99) and quality of life (MD 2.45; 95% CI 1.51 to 3.39).[50] The embedded qualitative study confirmed that women felt empowered, supported and valued by the counsellor.[51] In contrast, a Peruvian RCT (with some bias concerns) of a 30 min counselling session with a resource card and external referral had no effect on women's safety behaviours, health, use of community resources.[35] A Kenyan RCT of up to three 30–35 min counselling sessions with resource card, safety planning and external referral reduced depression (MD=7.12; 95% CI 6.21 to 8.03) and re-exposure to IPV (MD=13.51; 95% CI 9.99 to 17.02).[47] Similarly, an Indian UBA evaluation of two or more 30–45 min psychosocial counselling sessions found that most women had increased awareness of and readiness to address VAW. Physical violence and health problems decreased.[29]

### Coping skills training

Two RCTs with high risk of bias evaluated more intensive training interventions and found mixed effects on behaviour and mixed and null effects on VAW and health.[52 55] An Iranian RCT of four 90 min group sessions reported a reduction in re-exposure to physical IPV (RR 0.78; 95% CI 0.63 to 0.83) and psychological IPV (RR 0.73; 95% CI 0.64 to 0.83), but null effect on sexual IPV (RR 0.87; 95% CI 0.69 to 1.09).[55] A South African RCT of seven 90 min sessions reported null effect on coping behaviour, use of SRH services, post-traumatic stress disorder (PTSD) and HIV viral load among HIV positive women with a history of sexual violence.[52] However, the embedded qualitative evaluation found that training increased women's knowledge about VAW impact and improved their self-esteem, coping and communication skills.[44]

## Psychoeducation

An Iranian RCT with high risk of bias of four 90 min sessions with pregnant women reported reduction in scores of IPV and psychological distress.[40 41]

## Cost-effectiveness outcomes

No studies evaluated cost-effectiveness of VAW interventions in SRH services. One study of an integrated HIV-IPV consultation paid HCPs $6 per day for identifying patients experiencing VAW and referring them to the on-site GBV clinic.[57] One evaluation of post-rape service improvement with community engagement reported resource costs.[42 43] Seven studies across all three intervention categories mentioned intervention costs but did not report actual data.[30 38 47 50 53 56 57]

## Barriers to intervention effects and outcomes

Online supplemental file 7 summarises factors that women and HCPs perceived as barriers to intervention implementation and impact. We developed three analytical themes cross-cutting through individual, community, and system levels.

## Acceptability of VAW

Four evaluations of interventions on *response during routine SRH consultation* and *response with community engagement* described attitudes accepting violence and patriarchal gender norms as major barrier to behaviour change.[33 45 54 56]

## Fear of negative consequences

Eight studies across all three interventions categories identified fear of negative consequences as a barrier to identification, disclosure and engagement in VAW interventions.[30 33 38 44 45 49 51 53]

## Limited readiness

Evaluations reported limited readiness for engaging in VAW interventions at system and individual levels. In evaluations of *response to VAW during routine SRH consultation*[28 33 38 45 57] and *response with community engagement*,[30 49] HCPs consistently mentioned chronic problems with staffing, inadequate funding, no private space, lack of support from leadership and high demand for basic SRH services without the additional VAW work. Readiness gaps at system level included the lack of services to refer to, poor referral systems and untrained staff in other agencies. Screening identified many IPV-positive women and specialist services could not address the increased demand.[30 33 45 56] Across all intervention categories, HCPs and women described barriers at societal level that prevented women from accessing SRH services, using referrals and participating in psychosocial interventions. Work-related conflicts, no money for transport and financial dependence on husband were mentioned most frequently.[38 44 45 53] Finally, two evaluations of *response to VAW during routine SRH consultation* explored reasons for low uptake of referrals to specialist services. Some women had expectations that could not be met by the current services. Instead of referral, they wanted HCPs to talk to their partners about stopping the abuse.[33] Some women wanted to receive all SRH and VAW services on the same day.[38 57]

## DISCUSSION

We conducted a mixed-methods systematic review of studies from LMICs on the effectiveness and barriers to strengthening SRH services response to VAW. We grouped 26 heterogeneous complex interventions into three categories: (i) response to VAW during routine SRH consultation, (ii) response to VAW during routine SRH consultation plus community engagement and (iii) response to VAW in addition to routine SRH consultation. We mapped outcomes on a process-oriented logic model illustrating the hypothesised changes from intervention through direct and intermediate effects on health-related cognition, emotions and behaviour to health outcomes. We cannot conclude which intervention was the most effective in improving any of these effects and outcomes due to heterogeneity of the interventions and measures at varying time points. Overall, ten interventions did not escalate violence and two reported no harmful events. We found mixed effects on women's health and re-occurrence of VAW across all three categories, with studies reporting conflicting findings. Evaluations of the varied *responses to VAW during routine SRH consultation* found reduction in HIV-disclosure-related IPV and potential risk of becoming a victim of femicide, but no effect on IPV in the past month. Some of these effects could be attributed to improvement in HCPs' readiness, screening and provision of first-line support for VAW. For women, these effects could be attributed to increased knowledge about VAW and disclosure of violence. *Response to VAW during routine SRH consultation plus community engagement* had uncertain evidence with single studies reporting improvement, mixed effect, and no effect on re-exposure to violence and SRH possibly through some improvement in provision and use of SRH services. More intensive psychosocial interventions delivered to women with experience of VAW *in addition to routine SRH consultation* had the most robust evidence for reduction in re-occurrence of violence and improvement in health outcomes possibly through an improvement in safety behaviours. We identified individual, community and system-level barriers that could disrupt the pathway from intervention activities to outcomes: (i) attitudes and social norms that accept and normalise violence, (ii) fear of negative consequences and (iii) limited readiness of individuals, health systems and society to address VAW. No studies reported cost-effectiveness analysis.

## Strengths and limitations

This review is methodologically strong. It involved a multidisciplinary team of researchers from LMICs and the UK with content and methodological expertise in health systems response to VAW and global health. We

followed the Cochrane method and included studies of any design reported in peer-reviewed and grey literature in any language with English abstract. This comprehensive approach ensured inclusion of the most relevant studies from the field and reduced the potential for bias/errors.

The evidence for VAW interventions in SRH settings is weak because of the methodological limitations of the primary studies and uncertain effectiveness of the interventions. Each study used differing operational definitions and outcomes measures, relied on self-report, and evaluated a different complex intervention. No studies measured contextual and implementation factors nor adjusted their analysis for those factors which could mediate the effect of the intervention on outcomes of individual HCPs and women, although those were explored in some qualitative evaluations. Because of the diverse complex interventions and outcomes measures we could not perform a meta-analysis. Our findings should be interpreted with caution because two-thirds of trials and all 13 quasi-experimental studies had high risk of bias.

The evidence we found is applicable to ANC and HIV services and depends on the intervention category. Interventions that strengthened capacity of HIV and ANC services to respond to VAW can increase identification and provision of first-line support to women experiencing violence which can lead to reduction in HIV-disclosure-related IPV, physical and sexual IPV, and the risk of becoming a victim of femicide. More intensive psychosocial interventions that strengthen women's readiness to cope with VAW can increase use of SRH services and safety behaviours, reduce re-exposure to IPV and DV, and improve health and quality of life. The positive effect of additional psychosocial interventions can be explained by their theoretical underpinning, higher dose of provider–patient contact, delivery by study personnel specialised in counselling and VAW, and samples of self-selected women who could be more motivated and ready for change. The first two intervention categories might appear less effective because few studies examined women's outcomes beyond the point of identification and first-line response. Our findings suggest that future VAW interventions should strengthen multi-level capacity across individual HCPs and women, SRH services, and communities.

The uncertain evidence for the two SRH consultation-based intervention categories is consistent with other evidence for a healthcare response to VAW in LMICs[15 62] and to VAW among pregnant women.[16 63 64] The evidence for the effectiveness of longer psychosocial interventions as an addition to routine SRH consultations is in line with a recent meta-analysis which found that psychosocial interventions in healthcare settings and communities in LMICs led to a 25%–27% reduction in IPV.[65]

An important finding on the direct effects of interventions is that increasing awareness about VAW and relevant procedures often did not lead to a shift in judgemental attitudes towards victims, the major barrier to changes in professional and patient behaviour and practices. An exception was studies in the *plus community engagement category*, one reporting improvement in HCP attitudes and one partial shift in women's attitudes and gender norms. These findings can be explained by the community and societal roots of gender norms and attitudes of HCPs and women which are best addressed at community and societal levels. Our findings suggest that a shift in individual's attitudes potentially leading to behavioural change can be achieved through service-based plus community-based education. This finding is consistent with a review of evidence on what works to prevent VAW in LMICs. The review found good evidence for community activism approaches to shift harmful gender attitudes, roles and social norms.[66]

An important finding is that routine integrated SRH-VAW consultations with referral or signposting to VAW/GBV specialist or other services did not increase women's use of these services. This gap between availability and acceptability of referrals to other services has several explanations. Our themes of barriers to intervention impact suggested that HCP response may not have matched women's needs and expectations, or the VAW services were not accessible, or contextual factors prevented women from accessing them. This finding suggests that integrated SRH-VAW consultation and VAW services require better tailoring to women's needs and expectations. This should be based on understanding what women want and need and what is feasible. A recent qualitative meta-synthesis found that after disclosing IPV to HCPs, women wanted assistance with documentation of injuries, insurance issues and help with connecting to community services more than referrals to IPV services.[67] The feasibility and acceptability of HCPs engaging with men who use violence needs further exploration in LMICs.

Finally, most interventions in the first two categories targeted the behaviour of individual HCPs rather than the SRH service or health system. The expectation was that trained HCPs would integrate VAW work into their clinical practice without structural changes to the environment, support from leadership, supervision, monitoring and incentivisation. Most qualitative evaluations described passionate HCPs who were enthusiastic about helping patients experiencing violence. However, some studies reported HCPs concerns about unrealistic expectations and limited health system readiness for embedding VAW work in routine practice. This finding is in line with other studies on health systems readiness for responding to VAW.[9 10] The obstacles to integrating a VAW response in SRH services overlap with those reported in the systematic review of barriers and facilitators to integrating health systems responses to IPV in LMICs.[68]

This review's results are relevant for practitioners and policy makers in LMICs. The logic model approach allowed us to: (i) illustrate the hypothesised cause-result pathway, (ii) map evidence from primary studies for the direct and intermediate effects and outcomes, (iii)

identify barriers that can disrupt the trajectory of changes. It allowed us to present evidence in a format understandable to end users: people who develop, deliver, evaluate and fund VAW interventions in LMICs. We need more methodologically robust evaluations of interventions for strengthening the capacity of the health systems, communities and individual women to respond to VAW with measures throughout the pathway from intervention activities to women's outcomes. Absence of evidence on the cost-effectiveness of VAW interventions in SRH services is another gap. Finally, very few interventions have been evaluated in LMICs outside Africa.

## CONCLUSION

We found that interventions to improve response to VAW in SRH services did not escalate violence. Some interventions that strengthened capacity of HIV and ANC services increased identification and reduced some types of IPV. Some interventions that strengthened capacity of HIV and ANC services and communities improved use of SRH services and reduced re-exposure to some types of VAW. Several studies identified a gap between provision and uptake of referrals to VAW services suggesting that first-line support should be better tailored to women's needs and preferences. Most additional psychosocial interventions that strengthened women's readiness to respond to VAW reduced re-exposure to violence and improved health. Our findings are relevant to people who develop, implement, evaluate and fund VAW interventions in healthcare. Future interventions should have better theoretical development and use a systemic approach to strengthen the capacity to respond to VAW across the healthcare systems, communities and women. Future evaluations of VAW interventions in SRH services in LMICs should have longer follow-up and use standardised measures of individual-level, organisation-level and system-level outcomes on the pathway from intervention to women's health.

**Author affiliations**
[1]Bristol Medical School (PHS), University of Bristol Faculty of Health Sciences, Bristol, UK
[2]Department of Psychiatry, Faculty of Medicine, University of Peradeniya, Peradeniya, Sri Lanka
[3]Department of Global Health and Development, London School of Hygiene and Tropical Medicine, London, UK
[4]Faculty of Medicine, University of São Paulo Institute of Biomedical Sciences, Sao Paulo, Brazil
[5]School of Medical Sciences, Kathmandu University, Kathmandu, Nepal
[6]Faculty of Medicine and Health Sciences, An-najah National University, Nablus, State of Palestine
[7]NIHR ARC West, University Hospitals Bristol NHS Foundation Trust, Bristol, UK
[8]Department of Reproductive Health and Research, Organisation mondiale de la Sante, Geneve, Switzerland

**Contributors** GF, LB, NVL planned the review. All coauthors contributed to the protocol development. AR constructed and ran searchers. NVL and MM screened titles and abstracts. NVL, MM, LB, MC, AFD screened full texts. NVL, MM, MC, AFD, SP, SS, TR, AS, PR, AA, AR, LB worked in pairs on data extraction, risk assessment.
NVL summarised the findings. NVL, MC, LB developed logic model. NVL wrote first draft of the manuscript. All coauthors contributed to a further two revisions and approved final manuscript. NVL is responsible for the overall content as guarantor.

**Funding** This research was funded by the National Institute for Health Research (NIHR) (17/63/125) using UK aid from the UK Government to support global health research. AR was funded by the NIHR Applied Research Collaboration West (NIHR ARC West).

**Disclaimer** The funder of the study had no role in study design, data collection, data analysis, data interpretation or writing of the report. The corresponding author had full access to all the data in the study and had final responsibility for the decision to submit for publication.

**Competing interests** None declared.

**Patient consent for publication** Not applicable.

**Ethics approval** This study does not involve human participants.

**Provenance and peer review** Not commissioned; externally peer reviewed.

**Data availability statement** All data relevant to the study are included in the article or uploaded as supplemental information. There are no primary data in this work.

**ORCID iDs**
Natalia V Lewis http://orcid.org/0000-0002-4839-6548
Gene S Feder http://orcid.org/0000-0002-7890-3926
Loraine J Bacchus http://orcid.org/0000-0002-9966-8208

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
