## [Reviewer comments · BMJ Open]

ARTICLE DETAILS

TITLE (PROVISIONAL)	INTERVENTIONS IN SEXUAL AND REPRODUCTIVE HEALTH SERVICES ADDRESSING VIOLENCE AGAINST WOMEN IN LOW- AND MIDDLE- INCOME COUNTRIES: A MIXED-METHODS SYSTEMATIC REVIEW
AUTHORS	Lewis, Natalia; Munas, Muzrif; Colombini, Manuela; D'Oliveira, AF; Pereira, Stephanie; Shrestha, Satya; Rajapakse, Thilini; Shaheen, Amira; Rishal, Poonam; Alkaiyat, Abdulsalam; Richards, Alison; Garcia-Moreno, Claudia; Feder, Gene; Bacchus, Loraine

VERSION 1 – REVIEW

REVIEWER	Lawry, Lynn Brigham and Women's Hospital
REVIEW RETURNED	01-Jun-2021

GENERAL COMMENTS	Please address why only VAW was studied given we know that violence against men is underreported and unaddressed. Shouldn't men (equity) also be entitled to first-line support tailored to their needs and expectations which are vastly different? If anything, the fact that this did not include men, should be listed as a limitation. Therefore, I would suggest the selection criteria needs to say males were excluded or it assumes that all gender based violence happens against women. Methodologically, this is sound, of high veracity and a vigorous review with one exception: What did the authors specifically define as "effective" interventions? What is an "effective" program (i.e what do those indicators look like) is it better mental health? Access to abortion services if needed? Trauma counseling? This needs to be defined clearly and results need to talk about specifically about each indicator that defines effectiveness in the body of the text. There is a great deal in the supplementary files but none of the most important indicators are mentioned in the text (e.g Supplementary table 4). Overall, the manuscript lacks enough detail for me to understand the successful outcomes and again what the evaluators/authors thought an effective intervention might be. Is it a checklist of care? The NGO community has many documents that describe this checklist and what is considered capacity. Table 2 could be supplemental and the more detailed and important tables (4 and 5) that are now supplemental could be in the manuscript.
---

	Jewkes et al have a body of literature on what prevents VAW and is not referenced or discussed well. I would urge some of her work be included in the discussion. See: WhatWorks to decide if what was defined as effective for this paper follows her results. Finally, addressing GBV after it has happened has little to do with effectiveness of preventing GBV and is the bigger issue for VAW. Prevention is key. I think this difference needs a nod in the paper given that aftercare should try to address prevention.
--	---

REVIEWER	Vives-Cases, C Alicante University
REVIEW RETURNED	06-Jun-2021

GENERAL COMMENTS	Congratulations for this amazing study! In my opinion, it addresses an important public health issue and its results add a relevant contribution to our knowledge about the effectiveness of SV-related services. Only a few details should be reinforced in order to improve its coherence and clarity. -The research questions on page 5 must be integrated in the general aim of the study. In this sense, you should re-write the aim of the abstract in order to make sure that you integrate all relevant concepts of your review: effectiveness, cost-effectiveness, barriers of effectiveness and methodological quality assesment. -I think the section related to barriers of effectiveness is the weakest of your manuscript. The included barriers are not well-justified as barriers of effectiveness. In my opinion, you should guided this section according to review this section according to the criteria of health services effectiveness identified by Tanahashi, T. "Health service coverage and its evaluation." Bulletin of the World Health Organization vol. 56,2 (1978): 295-303. You can use this reference to guide and reinforce the results and/or interpret them in discussion section.
--

REVIEWER	Gausman, Jewel Harvard TH Chan School of Public Health, Women and Health Initiative; Department of Global Health and Population
REVIEW RETURNED	23-Jun-2021

GENERAL COMMENTS	This manuscript presents the results of an ambitious systematic review of interventions focused on violence against women within the context of sexual and reproductive health service delivery. The manuscript is well-written and the results are presented clearly. I only have a few comments to suggest to the authors at this time to strengthen the manuscript.  • The introduction section is concise and well-written. Some background however could be provided about the Health Systems Wheel and the models of health system responses to VAW in order to orient the reader to its later use. • Authors could provide a definition of how they operationalize SRH services, given that there may be different ways to define and categorize such services. • The authors do a very thorough job of discussing the results in the discussion section, though it could benefit from some reorganization to improve the readability. For example, some paragraphs are not clearly organized. For example, the second and third paragraphs in the discussion section (page 21, lines 16-
--

	46) lack a clear structure and seem to be a catch-all where key results are reiterated rather than synthesis.  • The authors may want to consider structuring the discussion more clearly around the logic model to improve the readability, as the logic model nicely organizes the • The authors state at least twice that “An important finding is that response to VAW in SRH services did not escalate violence.” However, from the logic model, it appears that only 2 studies assessed this, thus it seems like more nuance needs to be communicated along with this result so as to not overstate the findings.
--	--

VERSION 1 – AUTHOR RESPONSE

Reviewers' comments	Authors' response	Amended text
Reviewer 1		
1.1. Please address why only VAW was studied given we know that violence against men is underreported and unaddressed. Shouldn't men (equity) also be entitled to first-line support tailored to their needs and expectations which are vastly different?	Thank you for the opportunity to explain why we focused our review on violence against women. In the opening paragraph, we described with references the lack of evidence for health systems response to male victims and perpetrators (Introduction, para 1). We finished introduction by explaining that our study addresses gaps in evidence and recommendations for research on healthcare response to violence against women in low- and middle-income countries identified in the earlier systematic reviews by Kirk and Sapkota and the 2013 WHO clinical and policy guidelines on responding to intimate partner violence and sexual violence against women (Introduction, para 4).	Although IPV against men is increasingly recognised within the context of both same sex and heterosexual relationships, understanding of male victimisation and its health consequences is still poorly understood⁴ and there is a dearth of primary research on the healthcare response to male victims and perpetrators.⁵ Systematic reviews^{16 17} and WHO guidelines¹⁸ found scant evidence from LMICs on effectiveness of VAW interventions in healthcare. This study addresses the gap by answering two questions: (i) What is the evidence for effectiveness and cost-effectiveness of interventions in SRH services that address VAW? (ii) What are the barriers to the effectiveness?
1.2. If anything, the fact that this did not include men, should be listed as a limitation. Therefore, I would suggest the selection criteria needs to say males were excluded or it assumes that all gender based violence happens against women.	Thank you for this suggestion. We have added male recipients of healthcare services to exclusion criteria (Table 1).	Exclusion criteria: Male recipients of healthcare services.
1.3. Methodologically, this is sound, of high veracity and a	We are grateful for the positive feedback and the opportunity to define what we	We judged intervention effectiveness by improvement in any primary or secondary

Reviewers' comments	Authors' response	Amended text
vigorous review with one exception: What did the authors specifically define as “effective” interventions? What is an “effective” program (i.e what do those indicators look like) is it better mental health? Access to abortion services if needed? Trauma counseling? This needs to be defined clearly and results need to talk about specifically about each indicator that defines effectiveness in the body of the text. There is a great deal in the supplementary files but none of the most important indicators are mentioned in the text (e.g Supplementary table 4).	mean by an “effective” intervention. In data analysis, we clarified that we judged intervention effectiveness by improvement in any primary or secondary outcome listed in the individual studies (Data analysis, para 1). We have listed examples of primary outcomes in Table 1.	outcome listed in the individual studies (see Table 1). Table 1: Outcome is an event or measurement collected for participants in a study. Primary outcomes: any health outcomes for survivors of VAW (for example, re-exposure to VAW, sexual and reproductive health, mental health, physical healthy, quality of life), any harms, cost effectiveness of VAW interventions. AND/OR Secondary outcomes: patient and provider health-related cognitive and emotional outcomes (for example, knowledge, attitudes, confidence, readiness); health-related behaviour and practices (for example, identification and disclosure of VAW, provision and uptake of referrals and SRH services).
1.4. Overall, the manuscript lacks enough detail for me to understand the successful outcomes and again what the evaluators/authors thought an effective intervention might be. Is it a checklist of care? The NGO community has many documents that describe this checklist and what is considered capacity.	Thank you for the opportunity to clarify this. We did not use any checklists of care. In data analysis, we clarified that we judged intervention effectiveness by its impact on any quantitative health-related outcome as reported by authors (Data analysis, para 1). We provided our definitions of the health systems capacity and individual women capacity to respond to VAW (Methods, para 1).	We judged intervention effectiveness by improvement in any primary or secondary outcome listed in the individual studies (see Table 1). Our analysis was informed by the WHO Health Systems Wheel framework for responding to VAW.⁸ We defined the health systems capacity to respond to VAW as the cumulative availability and strength of the Health Systems Wheel building blocks. We looked at the capacity of the health systems at three levels: individual providers (e.g., knowledge, skills, practices), services and organisations (e.g., infrastructure, availability of supplies/medicines), community (knowledge, attitudes).¹³ We defined women’s capacity to respond to VAW as their readiness and ability to seek help, disclose abuse, get referrals, and receive services.
1.5. Table 2 could be supplemental and the more detailed and	Thank you for this suggestion, although incorporating it would make it difficult to	

Reviewers' comments	Authors' response	Amended text
important tables (4 and 5) that are now supplemental could be in the manuscript.	comply with the BMJ Open submission guideline and PRISMA statement (Page 2020). The submission guideline recommends up to five figures and tables. Tables longer than 2 pages will be published as online only supplementary material. On editor's request, we incorporated three tables under 2-page limit in the main document: Tab.1. Inclusion/exclusion criteria Tab. 2. Intervention components. Tab. 3. Integrated outcomes across all three intervention categories.	
1.6. Jewkes et al have a body of literature on what prevents VAW and is not referenced or discussed well. I would urge some of her work be included in the discussion. See: WhatWorks to decide if what was defined as effective for this paper follows her results.	Thank you for this suggestion. We referenced Jewkes's work (Kerr-Wilson et al, 2020) in the Discussion (para 7). However, the What Works programme of research focuses on the primary prevention of VAW in which the health systems only have a limited role. Our review focuses on the health systems response to VAW which contributes towards secondary and tertiary prevention, while Jewkes and al focuses primarily on primary prevention interventions at community level (rather than in healthcare). When responding to comment 1.7, we clarified the health systems roles in primary, secondary and tertiary prevention of VAW (Introduction, para 2).	This finding is consistent with a review of evidence on what works to prevent VAW in LMICs. The review found good evidence for community activism approaches to shift harmful gender attitudes, roles and social norms.
1.7. Finally, addressing GBV after it has happened has little to do with effectiveness of preventing GBV and is the bigger issue for VAW. Prevention is key. I think this difference needs a nod in the paper given that aftercare should try to address prevention.	Thank you for highlighting these issues. We appreciate the importance of preventing gender-based violence and recognise differing terminology used to define primary, secondary, and tertiary prevention of VAW. Our revision of the Introduction has been guided by the seminal text on the health systems response to VAW by Garcia-Moreno et al (2015) which sees prevention as a continuum (Introduction, para 2).	The healthcare system has a key role in preventing VAW because most women attend sexual and reproductive health (SRH) services at some point.^{6 7} The main role of the healthcare system is to contribute towards secondary and tertiary prevention through early detection of VAW and mitigation of its impact which can prevent ill health and reoccurrence of violence. Healthcare providers (HCPs) are uniquely placed to identify victims/survivors, provide first-line support and clinical care, and

Reviewers' comments	Authors' response	Amended text
		connect them with other services. Healthcare systems can also contribute to primary prevention through early identification of children exposed to violence in the home and support to programmes like home visiting or early childhood development. ⁸
Reviewer 2		
2.1. The research questions on page 5 must be integrated in the general aim of the study. In this sense, you should re-write the aim of the abstract in order to make sure that you integrate all relevant concepts of your review: effectiveness, cost-effectiveness, barriers of effectiveness and methodological quality assessment.	Thank you for this suggestion. We have re-written the abstract aim as recommended.	Objectives. To synthesise evidence on the effectiveness, cost-effectiveness, and barriers to responding to violence against women (VAW) in sexual and reproductive health (SRH) services in low- and middle-income countries (LMICs).
2.2. I think the section related to barriers of effectiveness is the weakest of your manuscript. The included barriers are not well-justified as barriers of effectiveness. In my opinion, you should guided this section according to review this section according to the criteria of health services effectiveness identified by Tanahashi, T. "Health service coverage and its evaluation." Bulletin of the World Health Organization vol. 56,2 (1978): 295-303. You can use this reference to guide and reinforce the results and/or interpret them in discussion section.	Thank you for an opportunity to strengthen the section about barriers and for signposting the Tanahashi paper. We have used the Health Systems Wheel framework for categorising barriers. Unfortunately, the 4000-word limit did not allow us to describe the themes in more detail in the text. That is why we submitted table 7 with descriptions of themes and supporting quotes. We developed themes of barriers from the analysis of qualitative studies. They summarise individual women and HCP views on the factors that they perceived as barriers to interventions implementation and impact.	
Reviewer 3		
3.1. The introduction section is concise and well-written. Some background however could be provided about the Health Systems Wheel and the models of health system responses to VAW in order to orient the reader to its later use.	Thank you for the opportunity to provide more detailed information in the Introduction. We expanded on the building blocks in the Health Systems Wheel and the models of health system response to VAW and provided seminal references (Introduction, para 2).	The Health Systems Wheel ¹¹ highlights key components that need to be in place to support individual HCPs and organisations to offer a comprehensive and client-centred response to VAW. It assumes that all elements of the health system – individual, organisational, contextual, and structural – impact on provision of response to VAW.

Reviewers' comments	Authors' response	Amended text
		The WHO guidelines for evidence-based health systems response to VAW adopted the Health Systems Wheel framework to recommend intervention activities across the health systems building blocks.¹²⁻¹⁴ In LMICs, healthcare delivery for VAW has been implemented through integration at the level of individual HCPs, healthcare facility, and healthcare system.¹⁵
3.2. Authors could provide a definition of how they operationalize SRH services, given that there may be different ways to define and categorize such services.	Thank you for this suggestion. We defined sexual and reproductive health services in Table 1 (Study inclusion and exclusion criteria with justification). For context, we explained: "Depending on country context, SRH services can be delivered at any level of healthcare provision and usually include contraceptive services, maternal and perinatal health, treatment for STI, HIV and reproductive tract infections, abortion, fertility treatment and gynaecological treatment."	
3.3. The authors do a very thorough job of discussing the results in the discussion section, though it could benefit from some reorganization to improve the readability. For example, some paragraphs are not clearly organized. For example, the second and third paragraphs in the discussion section (page 21, lines 16-46) lack a clear structure and seem to be a catch-all where key results are reiterated rather than synthesis.	Thank you for an opportunity to strengthen our Discussion section. We edited the second paragraph to elaborate on the meaning and possible explanations of our findings.	The evidence we found is applicable to HIV and ANC services and depends on the intervention category.
3.4. The authors may want to consider structuring the discussion more clearly around the logic model to improve the readability, as the logic model nicely organizes the	Thank you for this suggestion. We reformatted our Discussion section to adhere to the BMJ Open submission guideline (https://bmjopen.bmj.com/pages/authors/): statement of the principal findings (para 1); strength and limitations of the study (para 2-3), the meaning of the study and possible explanations in relation to other studies (para 4-8), practical implications and future research (para 10).	

Reviewers' comments	Authors' response	Amended text
3.5. The authors state at least twice that “An important finding is that response to VAW in SRH services did not escalate violence.” However, from the logic model, it appears that only 2 studies assessed this, thus it seems like more nuance needs to be communicated along with this result so as to not overstate the findings.	Thank you for an opportunity to explain the difference between harm and re-exposure to VAW and clarify how we concluded that most interventions did not escalate violence. In table 1 inclusion and exclusion criteria/outcomes, we separated re-exposure to VAW and any harms, because the latter includes any adverse events related to a VAW intervention. Only two studies explicitly reported whether engagement with a VAW intervention resulted in any harmful effects. We reported this finding as a separate health outcome. In total, 10 studies measured re-exposure to VAW and found reduction (n=6), mixed effect (change in some types and nil effect on other types, n=2), or no change (n=2). This allowed us to conclude that although some intervention did not change re-exposure to VAW, none resulted in violence escalation (Results/Intervention effects and outcomes/Health outcomes).	Only half studies reported measures of health and re-exposure to VAW; 2 interventions reported no harm resulting from taking part in interventions,^{35 38} and 10 led to some health improvement. We found that although some interventions did not reduce re-exposure to VAW, none reported violence escalation. Of 10 studies that measured re-exposure to VAW, 6 found a reduction,^{35 37 41 42 51 54} 2 reductions in some violence types but not in others,^{50 52} and 2 reported no change.^{38 44}